# Risk and Protective Factors of Well-Being among Healthcare Staff. A Thematic Analysis

**DOI:** 10.3390/ijerph17186651

**Published:** 2020-09-12

**Authors:** Sabrina Berlanda, Federica de Cordova, Marta Fraizzoli, Monica Pedrazza

**Affiliations:** Department of Human Sciences, University of Verona, 37129 Verona, Italy; federica.decordova@univr.it (F.d.C.); marta.fraizzoli@univr.it (M.F.); monica.pedrazza@univr.it (M.P.)

**Keywords:** well-being, educators in residential care, nurses, thematic analysis

## Abstract

The purpose of this study was to identify physical and psychosocial working conditions to improve well-being at work among healthcare staff. This is a potent area of inquiry given the relationship between healthcare staff well-being and service quality and other key organizational characteristics. However, while numerous studies in this area have used a quantitative methodology, very few have applied qualitative methodologies gathering subjective descriptions of the sources of well-being, providing in so doing significant data to explore in depth the factors that influence well-being in healthcare systems. We gathered qualitative data analyzing open-ended questions about risk and protective factors of well-being at work. The sample was made of 795 professionals answering an online questionnaire. Answers were coded and analyzed using the thematic analysis with an inductive approach (data-driven). We identified four themes strongly affecting professional well-being in health-care staff: *Interactions*, *Working Conditions*, *Emotional Responses to Work*, and *Competence and Professional Growth*. Our findings suggest possible strategies and actions that may be effective in helping to calibrate case-specific support and monitoring interventions to improve health and well-being of healthcare staff. We also discuss the implications of the study and suggest possible avenues for future empirical research.

## 1. Introduction

The creation and implementation of effective healthcare systems is generally regarded as an essential step in a country’s development [1]. Healthcare staff is an umbrella term involving a wide array of professional roles engaged in actions whose primary intent is to enhance health. They treat and prevent human illness and other physical and mental difficulties in accordance with the needs of their communities [2]. Most of the literature, involving healthcare staff, has focused on medical professionals [3,4,5]. This has resulted in a relative underrepresentation of other groups of healthcare staff [6,7,8].

In this contribution our aim is to highlight the role of two different groups of frontline health workers, that provide routine and care services to users of healthcare systems: ward nurses and educators working in residential care centers. Specifically, residential care centers for child protection, and residential care centers for disabled children. In the Italian system, this professional role is at the border of multiple skills and abilities and their background involves social, pedagogical, healthcare knowledge. Educators play a connection role among different specialists (social workers, psychologists, child psychiatrists, and neuropsychiatrists, etc.) and between the life in the community and the outside world (school, family etc.). Their task is to structure daily routines in order to apply a multidisciplinary project of care, organizing effectively social and healthcare resources, in order to enhance subjective potentialities and to increase personal autonomy. Both these professionals are considered the “frontline” [9,10], having the most direct and recurrent interaction with the users [11]. They play a critical role, owing to their greater access to individuals and families and the considerable time they spend with them to take care of them [12]. Both these professionals share the relational field of the dyadic relationships with users of healthcare systems as the context where they are required to exhibit their professionalism and competence [13]. We therefore assume that despite the very different work environment and work practices, they should share common challenges and criticalities and hence exhibit a very similar profile relating to well-being at work.

Educators and ward nurses both work in complex and challenging environments and climates, sharing the common objective of reconciling two sometimes divergent work demands. On the one hand, they have to carry out activities and functions which are widely integrated and structured in the institutional systems. They have in fact to exert a certain amount of control over users in order to integrate them virtuously in the healthcare system. On the other hand, they have to establish an alliance with the user or the patient [14,15]. They also have to support and encourage users to trust more in themselves and to develop hope and autonomy. Both perspectives, the more structured and control-oriented one and the caring and supporting one, need to be considered.

### 1.1. The Key Role of Health Care Professionals

The centrality of relationship exposes educators to the immediate users’ needs. In doing so, they are the frontline staff, the first to cope with users’ crises, anger, aggression, and interpersonal conflict resolution [16]. Educators need to regulate their own emotions in order to work effectively and productively [17]. Emotional labor is highly energy consuming and sometimes the organizational, interpersonal, and intrinsic characteristics of the work environment can fail to sustain the employee. The result can be disruptive for the individual, ending up with job stress, burnout, and intent to leave [18]. Nurses constitute the most numerous health-care profession [19]. The World Health Organization has identified nurses as a key human resource in health care and has highlighted the need to strengthen the nursing workforce to improve health outcomes [20,21]. Nurses play a pivotal role in improving patient outcomes. They are often the only health professionals that people have access to [22], and are the principal healthcare professionals in hospitals, providing patients and their families with holistic care 24 h a day [10]. In fact, in hospitals, nurses effectively constitute an around-the-clock surveillance system for early detection and prompt intervention when a patient’s condition deteriorates [23]. This kind of relationship requires continuous and prolonged nurse–patient interaction, in which a significant amount of time is dedicated to exploring the perspectives and personal needs of patients. Given the above-mentioned characteristics, the work of both nurses and educators in residential care is usually more than “just a job,” so it is important to ascertain personal meaning from one’s contact with patients [13,24].

### 1.2. Well-Being at Work

In 2014, a joint report from the European Foundation for the Improvement on Living and Working Conditions and the European Agency for Safety and Health at Work revealed that managers in the field of health, social work, and education were the most concerned about the psychosocial risk of their employees, although concern is not automatically translated into tools to face the risk and manage it [25]. Today, improving employees’ well-being is a central element of any organization’s “mission”, and measuring staff well-being is a key objective for virtually all institutions and companies [26]. To make this possible, it is important to have a clear, transparent, and consistent understanding of what well-being means.

Low levels of well-being have been closely associated with lower job commitment and productivity, occupational stress, absenteeism, turnover, and the delivery of poorer-quality care [8,13,23,27,28]. Poor healthcare staff well-being also has a significant economic impact [8,29]. On the contrary, satisfied and happy workers are more productive [19,30,31], and perform better [26]. Healthcare staff who are satisfied with their jobs are usually more confident about their ability to do the job and enjoy an overall sense of capability in relation to the organization and delivery of specific activities, even when these are not, in themselves, entirely successful [14]. Healthcare staff well-being also play an important role in patient safety [27,32]. Given that well-being has been found to be related to performance in the work setting, it is not surprising that the concept of well-being has attracted much attention [1,8,27,33].

Staff well-being in healthcare is a complex topic, in order to tackle such complexity, it is worthwhile to give voice to the person directly involved in the process of care. A review study explores how healthcare staff perceive and experience the care and provides relevant insights to promote high quality healthcare services in a specific hospital service [34]. For this reason, we applied qualitative methodologies to gather subjective descriptions of well-being in a wider healthcare staff, considering such approach appropriate to provide significant data to explore in depth the factors that influence well-being in healthcare systems.

### 1.3. Risk and Protective Factors Affecting Well-Being at Work

Health and well-being among healthcare staff are affected by many factors. Literature distinguishes between risk and protective factors of well-being. A risk factor is a characteristic, condition, or behavior that increases the likelihood of getting a disease or injury. Risk factors often coexist and influence one another, they are presented individually, however in practice they do not occur alone. Every healthcare institution faces the challenge of maintaining well-being, especially in times of economic and societal strain, which lead, for instance, to continual budget cuts [35]. Attempts to contain costs can involve additional paperwork, an increased focus on the routinization of service. Initiatives of this sort in health care may lead to a more efficient performance at lower costs. However, they also increase levels of stress for healthcare staff, leading to decreased well-being and a negative work environment [36]. In recent years, healthcare staff has become an urgent concern that is related to the stresses placed on society as a whole: the situation is compounded by surging population growth and by greater service delivery demands [37]. This situation leads professionals to perceive a poor-to-average quality of patient/service-user care [23,37], and increased levels of stress and burnout [38,39].

In the contemporary society there are many issues for the educator to be mastered, like job insecurity, poor contract terms, ever-changing working conditions and related skill building, high level of turnover, short term goals, and technological innovation. All these difficulties are highly demanding for this category of professionals, as much as the fact that the organizational setting is vanishing as a resource to sustain their professional attitudes and behaviors. Under these circumstances, well-being can be hard to experience, and isolating their precursors is essential to develop healthy and effective work environments. The profession of “nurse” has similar problems. The literature identifies two problems that particularly affect the Western health systems: turnover and shortage of nurses [40,41,42,43,44]. The World Health Organization explored the negative impact of human-resources shortages on global health care [45]. Over the last two decades, the nursing profession has experienced a drop in numbers caused by a combination of fewer people entering the profession, a crisis in retention—the result of many qualified nurses leaving because of difficult working conditions [46,47], unrewarding work environments [14,48]—and an ageing workforce [49]. The shortage problem is a vicious circle because the low number of nurses means those who are employed face a greater workload [50], which in turn causes stress and dissatisfaction with work, ultimately leading more nurses to leave the profession, further decreasing their number [41]. The nurse shortage is an important issue because it affects patient care by limiting how much time nurses can spend actively collaborating with team members and attending to individual patients. Both these factors have implications on patient safety as well as the nurses’ capacity to detect complications early [51]. In conclusion, burnout and turnover among healthcare staff are not just an organizational challenge or a topic for economic analysis but a global issue with real-world effects in terms of quality of health care [23,52,53].

Protective factors of well-being are individual or environmental characteristics, conditions, or behaviors that reduce the effects of the above-mentioned risk factors. Protective factors that enhance healthcare staff well-being have been found in an Italian study to include job satisfaction and manageable workloads, this being moderated by realistic task expectations, professional values, and the degree to which there is a sense of community within the multidisciplinary team [13,54]. Different authors have defined job satisfaction differently, but there is a consensus that job satisfaction is related to a positive attitude towards the individual’s job [42,55]. Job satisfaction has been associated with professional, personal, interpersonal, and organizational variables [14,56] and is considered a multi-dimensional concept [19,26], in which each dimension is important [26,36,57,58,59].

## 2. Materials and Methods

Healthcare staff bear a great amount of responsibility and face high expectations. To ensure the quality of healthcare systems, it is important that we have a good understanding of the experiences of these professionals and, particularly, of how they perceive their work, and of the different factors that determine their well-being at work. We frame well-being as a complex, multilayered construct: some factors can improve levels of well-being, while others work as risk factors against it [27]. The aim of our study is to explore healthcare staff’s perceptions and identify and clarify their sources of well-being at work. Our research question was as follows: how do people directly involved in the healthcare work confer meaning to “source of well-being at work” and “risk for well-being at work”? To answer this question, an online self-report questionnaire was administered including six open-ended questions on the topic; data were analyzed according to thematic analysis, more apt to deepen the subjective viewpoint of healthcare staff [60]. Thematic analysis is a flexible method not directly tied to a particular epistemological or theoretical perspective [61]. It highlights participants’ subjective experiences and sense-making focusing on the participants’ perceptions, feelings, and experiences and guarantees structural conditions that allow the emergence of relevant data [60]. Workers’ own perspectives and their perception of their experiences of well-being at work offer the most useful information for implementing change. Increased well-being could improve the quality of care and service provision in the healthcare sector.

### 2.1. Procedure

An online self-report questionnaire was administered to a convenience sample of educators working in non-profit organizations in northeast Italy, and to a convenience sample of registered ward nurses working in hospitals, also in northeast Italy. The participants’ email addresses were provided by the administrators of the non-profit organizations and the hospitals in question. Ethical approval was obtained from the Ethics Committee at the researchers’ institution (reference number 284-77324 on date 26 November 2014). The questionnaire included a section that explained the nature and the purpose of the study and a consent form. Participation was entirely voluntary and informed consent was obtained from each participant. Participants were informed of their right to withdraw or refuse to give information at any time without negative consequences. We took appropriate measures to ensure the privacy and anonymity of individuals involved in our research, including anonymizing their answers. The questionnaire included three open-ended questions about protective factors of well-being at work (“What is your first/second/third most important source of well-being at work?”), three open-ended questions about conditions at work that put healthcare staff at risk of poor physical and mental health and well-being (“What is your first/second/third most important risk factor of well-being at work?”), and some questions on demographic and occupational characteristics (gender, age, length of service).

### 2.2. Participants

In total, 1416 professionals were contacted by e-mail. A total of 795 questionnaires were completed, with a response rate of 56.14%. The sample was composed of 423 educators (53.2%), and 372 ward nurses (46.8%). A large majority of participants were female (605 of 795, 76.1%). In terms of age, the largest group of respondents (54.8%) were in the 31–50 age range, 23.4% were in the 21–30 range, 19.0% were in the 51–60 range, and 2.3% were older than 60. Age data were missing for five respondents (0.6%). About half of the sample (46.0%) had up to 10 years of experience in their profession, 33.4% had between 11 and 20 years of experience, and 19.6% had more than 20 years of experience (data missing for eight respondents, or 1.0%). Gender, age, and length-of-service data for healthcare staff are reported in Table 1. The sample of educators was composed of more male than the sample of nurses (educators 28.1% and nurses 19.1%). Moreover, in terms of age and length-of-service, educators were younger than nurses (30.4% of educators are under 30 years old) and they had fewer years of experience (60% of educators have a seniority of less than 10 years).

### 2.3. Data Analysis

Qualitative analysis was performed with NVivo 11. Open-ended questions allow to give voice to the subjects, discussing the topic in their own words, providing significant insights for the research. A multi-stage standardized thematic analysis was used to code and interpret the open-end answers [61,62,63,64]. Thematic analysis is a useful and flexible research instrument for identifying and analyzing patterns in qualitative data [61]. It “captures something important about the data in relation to the research question and represents some level of patterned response or meaning within the data set” [63] (p. 82). By reading the answers carefully, recursively, data items relevant for the research question are gathered together and coded according to labels well-being/risk. In this way themes are identified according to patterns of meaning.

Using a semantic approach, we identified themes within the explicit meanings of the data. Our analysis was not looking for anything beyond what the participant had written. We adopted an essentialist/realist approach, whereby we assume that what the participants say and how they express it actually reflects their experience. Two researchers (the first and third authors) independently analyzed narratives of the open-ended answers. Themes were identified in an inductive or “bottom-up” way. As such, thematic analysis of this sort is data driven [61]. We also adopted a recursive process [62]. The first step of the interpretation process was open. In this first level of data analysis the data were read actively and researchers identified initial codes, salient themes, recurring ideas or language, identifying of the most basic meaningful semantic elements of the data. Guided by inductive reasoning, codes were assembled based on similarity, and similar codes were combined to sub-themes. In the second step of the interpretation process data were organized and summarized and categories were drawn up and combined to form overarching themes. At this point, the potential themes and sub-themes were reviewed to assess their internal homogeneity and external heterogeneity. Finally, the themes were refined and defined, and the data analyzed. The two researchers independently detected regularities and generated categories ad hoc for each theme via an inductive process (key words or abbreviations). Cohen’s kappa was 0.82, which indicates a good inter-rater reliability [65]. The attempt in this analysis is to provide narrative samples of healthcare staff’s own perspectives and their perception of their experiences of well-being as they arose unprompted in the open-ended answers.

Mann–Whitney tests (a nonparametric analysis) were performed with SPSS (Version 21.0, IBM, Armonk, NY, USA) to explore differences between the two groups—educators and nurses—in terms of the themes and sub-themes that emerged in their answers. We also explored the differences between educators and nurses based on their age (21–40 years old educators versus 21–40 years old nurses; over 40 years old educators versus over 40 years old nurses).

## 3. Results

### 3.1. Word Frequency

We used word frequency queries in NVivo 11 to list the words that occurred most frequently in answers to open-ended questions. These queries allowed us to analyze the words used most frequently by educators and nurses when describing their risk and protective factors of well-being at work. Results are reported as word clouds (Figure 1 and Figure 2). These clouds display up to 60 words in different font sizes. The more frequently a word occurs, the larger the font used to display it. Similar words are grouped into single “terms” (for example “thinking” includes “think” and “thinks”). Figure 1 shows the words used by healthcare staff when asked to indicate three protective factors of well-being, while Figure 2 shows the words used by the same groups to indicate three risk factors for their physical and mental health. Relationships with colleagues, and relationships in general, seem to play a crucial role in affecting the professionals’ physical and mental health: therefore, relationships among colleagues are both a risk and a protective factor. However, it is worth noting that the relationship with users of healthcare systems occupies a central and strategic position as source of healthy working conditions for healthcare staff. The words that described specifically well-being factors are “colleagues”, “equipe”, and “collaboration”; while the words that occurred most frequently for risk factors at work are “workoverload”, “recognition”, “shift”, “difficulty”, and “time”.

### 3.2. Thematic Analysis

Thematic analysis identifies the most relevant risk and protective factors of well-being at work, which are brought together under four overarching themes (Figure 3): *Interactions* (35.60%; educators 33.00% and nurses 38.50%), *Working Conditions* (25.70%; educators 24.97% and nurses 26.50%), *Emotional Responses to Work* (21.99%; educators 20.97% and nurses 23.10%), and *Competence and Professional Growth* (16.71%; educators 21.06% and nurses 11.80%).

The *Interactions* theme comprises two sub-themes (Figure 4): *Interactions with Colleagues and Supervisors* (57.91%), and *Interactions with users and their families* (42.09%). Relational factors that the healthcare staff identified as risk and protective factors of well-being at work include: situations of support or conflict involving colleagues and supervisors (e.g., “I trust my colleagues”; “My supervisor considers my point of view”; “I don’t feel to be part of a team”; “My supervisor doesn’t promote a fair working environment”); cooperative or obstructive behaviors on the part of users and positive collaboration or problems with families (e.g., “When I’m able to build trust with users”; “I can play a supportive role for the user’s family”; “Patients making claims”; “Relational problems and communication issues with the families”). Mann–Whitney tests were conducted to explore paired differences between educators and nurses (Table 2). The results show that nurses attribute greater importance to *Interactions* as sources of well-being than the sample of educators (educators 38.99% and nurses 49.30%; *p* < 0.020). Furthermore, nurses assign more importance than educators to poor or negative interactions with users as sources of poor physical and mental health and well-being (educators 30.47% and nurses 33.58%; *p* < 0.030). This datum is enhanced in age-related sub-samples (Table 3 and Table 4): younger nurses identify difficulties in *Interactions with users and their families* as a risk factor of well-being more than younger educators do (educators 32.34%, nurses 41.11%; *p* < 0.040). In any case, according to our sample *Interactions* is the main factor that influences well-being at work for healthcare staff.

The theme Working Conditions comprises four sub-themes (Figure 5): Control over Work Scheduling and Perceived Work Overload (37.87%), Work Organization (36.19%), Remuneration and Job Security (17.45%), and Physical Space and Tools (8.49%). Within this theme, healthcare staff identified the following as risk and protective factors of well-being at work: workload and opportunities to make decisions and to be creative (e.g., “On my job, I am given a lot of freedom to decide how I do my work”; “My workload has increased as a result of the lack of resources”); shift patterns and work-family balance (e.g., “Flexibility in working hours”; “I have the possibility of achieving a real work-life balance”); salary and remuneration (e.g., “My salary is adequate”; “I don’t make enough money”); the availability or otherwise of adequate space and tools (e.g., “The safety of the building where I work”; “A lack of effective tools for helping users”). The results show that nurses assign more importance than educators to Working Conditions as risk factors for health and well-being (educators 36.91% and nurses 43.29%; *p* < 0.030). Age-related sub-samples confirm the difference between the two groups of professionals: younger nurses, compared to younger educators, attribute greater importance to poor or negative working conditions as sources of poor physical and mental health and well-being (educators 37.64%, nurses 49.19%; *p* < 0.020). Working Conditions proves to be crucial and is considered the main workplace risk factor by our sample of healthcare staff.

The theme *Emotional Responses to Work* comprises three sub-themes (Figure 6): *Achievement and Self-Fulfillment* (36.10%), *Emotional Labor and Emotion Regulation* (35.98%), and *Social Recognition* (27.92%). Risk and protective factors of well-being at work identified by healthcare staff that were grouped in this theme concerned their feelings and emotional responses in relation to the following factors: self-fulfillment and self-actualization (e.g., “I do a job that satisfies me”; “I feel frustrated, it is not what I want to do in my life”); physical and psychological stressors related to the health and safety of users (e.g., “I’m happy to see that what I do improves the life of users”; “It is difficult to be in contact with the suffering.”); social recognition and occupational prestige (e.g., “In my work I have a high responsibility”; “I’m not often respected”). The results show that educators attribute greater importance to *Achievement and Self-Fulfillment* as sources of well-being than the sample of nurses (educators 75.44% and nurses 36.89%; *p* < 0.020). In contrast, nurses assign more importance to difficulties with emotion regulation, identifying this area as an important risk factor for well-being (educators 27.68% and nurses 52.61%; *p* < 0.030). If we look at the differences in age-related sub-samples the results show that older educators, compared to older nurses, attribute greater importance as protective factor of well-being to *Achievement and Self-Fulfillment* (educators 82.80%, nurses 70.59%; *p* < 0.040).

The theme *Competence and Professional Growth* comprises two sub-themes (Figure 7): *Perceptions of Work Ability* (58.11%) and *Professional Skills and Competencies* (41.89%). Within this theme healthcare staff identified the following as risk and protective factors of well-being at work: a perceived ability or lack of ability to deliver good quality service (e.g., “Experience teaches me new ways of intervention”; “I’d like to give a better service”); training activities and lifelong learning (e.g., “I always learn new things through training courses”; “I possess few technical competences”). The results show that the educators attribute greater importance to *Competence and Professional Growth* (educators 19.83% and nurses 7.54%; *p* < 0.020) and to *Perceptions of Work Ability* (educators 46.89% and nurses 26.03%; *p* < 0.050) as risk factors for well-being than the sample of nurses. Analyzing more in depth age-related sub-samples, results show that younger educators, compared to younger nurses, attribute greater importance as protective factor of well-being to *Competence and Professional Growth* (educators 30.00%, nurses 22.20%; *p* < 0.020). In contrast, younger nurses assign more importance to lack of ability to deliver good quality service, identifying this area as important risk factor for well-being (educators 70.49%, nurses 84.62%; *p* < 0.020).

## 4. Discussion

Promoting well-being among healthcare staff is not only in the interests of these professionals themselves but is also important for optimal standards of care and service delivery [27,38]. Risk and protective factors of well-being at work have a determinant role in healthcare staff’s achievement and motivation levels at work and have an impact not only on the well-being of the individuals involved but also on that of the organization or institution, and indeed that of society [66]. This study presents a qualitative examination of healthcare staff’s perceptions and experiences. Using thematic analysis, we identified the most relevant risk and protective factors of well-being at work for sample groups from these two professions.

The first theme in our list is *Interactions*. Several studies highlight that interpersonal relationships with users of healthcare systems, co-workers, and supervisors are an important factor to improve well-being (if they are cooperative and supportive in nature), and for poor physical and mental health (if they are characterized by tension and conflict) [19]. The *Interactions with colleagues and supervisors* sub-theme relates to how employees perceive their own specific work-related interactions. The quality of relationships differs from one institution to another because they are largely determined by the leadership style adopted or displayed by the managers in that specific case [67]. Supportive and caring leaders, and good-quality relationships between leaders and employees increase employee satisfaction and performance [21,68]. User-professional relationships provide health care staff with a sense of fulfillment and give meaning to their work [31]. In particular, for nurses, good relationships with doctors, nursing managers, colleagues, and patients are resources that can improve well-being and job satisfaction [69].

The second theme is *Working Conditions*. In the literature, the impact of poor organization, insufficient staffing, and work overload are shown to be associated with job dissatisfaction and fair or poor standards of service [19,49]. Our results confirm that remuneration is not the primary motivation for workers in these professions; these data are consistent with previous literature showing that nonmonetary work conditions and organizational climate are more important aspects for well-being at work [70]. 

The third theme is *Emotional Responses to Work*. The development of effective emotion-regulation skills is critical to the success of professionals in a variety of settings [71]. This competency is an essential component in the professional development of healthcare staff, who must learn to interact successfully every day with a diverse array of users of healthcare systems and their families, as well as with colleagues and superiors. The way emotion is managed can have a positive impact on the functioning of health services, but also has the potential to foster a sense of vulnerability, stress, or exhaustion in the healthcare staff [72,73]. Such stressors can initiate strong negative emotions that can interfere with professionals’ capacity to deliver high-quality service [71], and even have a negative effect on their health. Difficulties in managing negative emotions and general professional stress are frequently cited as two of the primary contributing factors in cases of dissatisfaction at work [74]. Social-emotional competencies and stress management are thus key capacities in promoting and maintaining well-being and performance in healthcare staff [68]. This theme is also concerned with the workers’ feelings about how they and their work are recognized in the wider society. Recognition from others is also known to have a positive effect on well-being and quality of service [31], and the chronic undervaluing of healthcare staff is a common problem all around the world [75]. In recent years, the meaning attached to the job of these professionals has changed, with roles that were previously an unvalued, domestic vocation now perceived as higher-status professions that require technical skills [74].

The last theme is *Competence and Professional Growth*. Personal growth, and the opportunity to help users of healthcare systems are important sources of pleasure and well-being at work [19,33]. Training and development are important in ensuring long-term career prospects [51], promoting job satisfaction, providing continued provision of quality service [19], and enabling workers to handle demanding work.

### 4.1. Risk and Protective Factors of Well-Being at Work for Healthcare Staff

Educators and nurses exhibit a similar well-being profile. The relational dimension of the work practices and their affective features constitute the core sources of well-being for healthcare staff. The affective dimension of the relationship exerts an important role in determining and supporting the professionals’ well-being, with specific reference to the fact that the users’ or patients’ well-being and satisfaction are associated to professionals’ self-realization purposes and to the social and public recognition of their positive role. Professional growth has been identified as the third source of well-being whereas the organizational variables occupy the last position in guaranteeing the well-being of these professionals.

Work overload and poor work organization have been identified as main risk factors of well-being for healthcare staff. This is congruent with up to date literature concerning stress related issues and dissatisfaction in helping professions. If precarious, or to some extent compromised, the emotional dimension of relationship enhances job dissatisfaction, for both educators and nurses. More specifically, whereas nurses compromise their satisfaction when faced with problematic terminally ill patients, severe physical suffering and the coping with death, educators are mainly concerned with frustration and lack of social recognition for their professional role, that latter also enhanced by the perceived very unsatisfying wage.

### 4.2. Implications and Strategies to Improve Well-Being at Work

To deliver quality patient care, the care must first and foremost be safe, and literature suggests that healthcare staff well-being may play an important role in patient safety [27,76,77]. It would seem prudent that healthcare organizations provide a work environment that fosters staff well-being and protects against risk factors, to subsequently provide a safe service to their patients. The direction of the relationship between healthcare staff well-being and the quality and safety of care is unclear and the two factors may operate as a feedback loop: higher staff well-being may lead to better quality and safety of care, but an inability to provide high quality, safe care may lead to disillusionment, stress, and poor well-being [76,78]. Literature emphasized the effectiveness and the positive uptake of interventions which target healthcare staff well-being and improve patient care together [76,79]. Our findings suggest four areas in which managers should intervene to increase healthcare staff well-being: *Interactions*, *Working Conditions*, *Emotional Responses to Work*, and *Competence and Professional Growth*. Interventions can focus both on supporting individual skills (person-directed interventions) and targeting the workplace with an organizational approach (organization-directed interventions) [76]. Actions tackling both the levels (person and organization-directed) are more effective for creating sustainable and high-quality health services [79,80]. Specialized training should be guaranteed in order to enhance interpersonal, emotional, and social competences in healthcare staff [34,81]. Participation in arts-based activities for example has been proved to exert effects on different levels, such as psychological, emotional, and social [82]. Professionals should be supported not only in facing typical and widespread burnout symptoms, but also to cope with cultural and social rapid transformations which often suggest new and different ways of working. To do so, healthcare staff should develop skills including stress management, team communication, and teamwork and be provided with psychological tools such as cognitive behavior therapy, psychological therapy, clinical supervision, mindfulness groups, or counselling.

On the other hand, organization-directed interventions include work scheduling changes, in order to improve staff–patient ratios, and in so doing providing staff with more time to monitor and care for patients. In addition, time scheduling should leave space for clinical supervision and training. Furthermore, interventions supporting staff well-being implies the involvement and engagement of the management and board-level staff, supporting their leadership skills, improving management capability so as to deliver strong visible leadership and support healthcare staff [76,79]. In fact, as literature suggests, service managers should note that supportive and caring leaders and the promotion of an authentic leader-follower relationship typically increase employees’ well-being and performance [83]. Last but not least, support for healthcare staff after incidents or exposure to workplace violence should be guaranteed in order to recover well-being [84,85].

### 4.3. Limitations and Future Directions

We recognize a number of limitations in our study. First, it is essentially cross-sectional in nature. Future longitudinal studies could explore the variability of risk and protective factors of well-being over time. Furthermore, our sample is a sample of convenience. We also rely exclusively on self-reports, and it is possible that participants may not be able or willing to articulate their perceptions and that some may provide what they consider are socially desirable responses. Finally, this study was conducted using a qualitative methodology and this may affect the generalizability of our results. Subsequent studies may wish to apply quantitative analysis methodologies to the four risk and protective factors of well-being at work identified in our project.

## 5. Conclusions

Staff well-being in healthcare is a complex topic suggesting a multifaceted underlying process. In order to tackle such complexity, this study provides qualitative data regarding subjective descriptions of sources of well-being in healthcare workers. Through a thematic analysis we highlighted the main risks and protective factors of workers’ well-being in healthcare, that is *Interactions*, *Working Conditions*, *Emotional Responses to Work*, and *Competence and Professional*, where *Interactions* is the main source of well-being, and *Work Conditions* is the main risk factor. Applying qualitative research methodologies to the topic gives voice to the person directly involved in the process of care and provides relevant insights to promote high quality healthcare services. 

## Figures and Tables

**Figure 1 ijerph-17-06651-f001:**
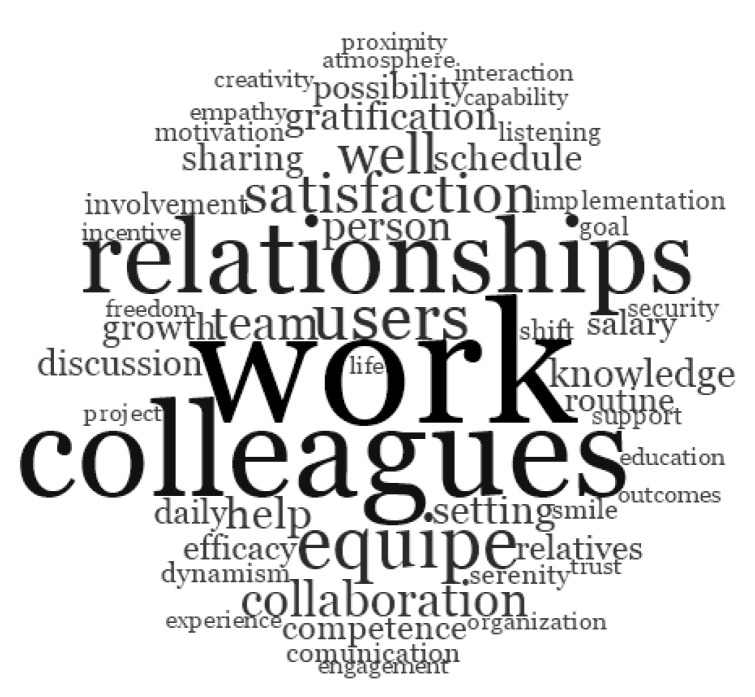
Words used to describe the three protective factors of well-being at work.

**Figure 2 ijerph-17-06651-f002:**
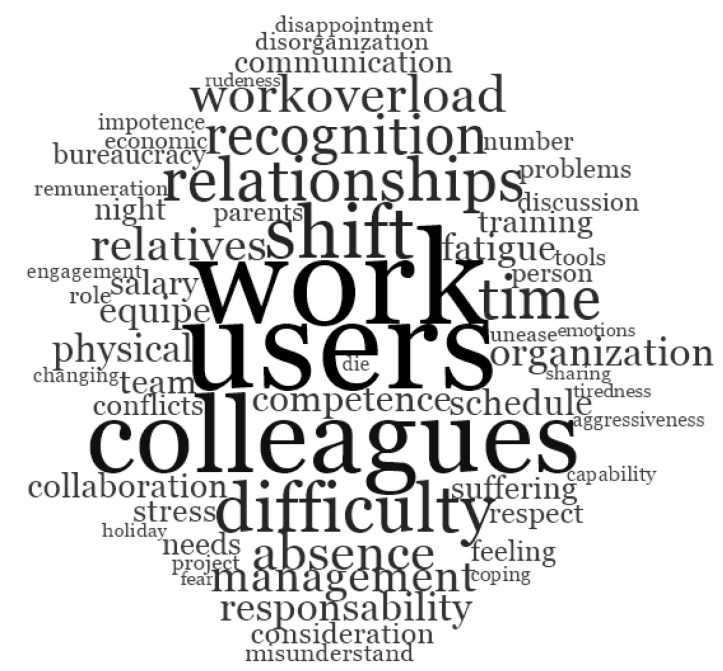
Words used to describe the three risk factors of well-being at work.

**Figure 3 ijerph-17-06651-f003:**
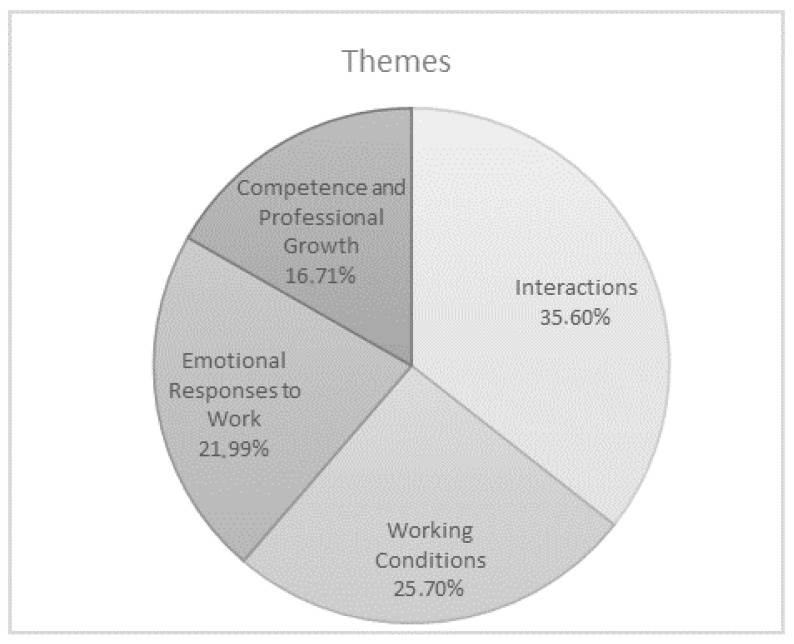
Risk and protective factors of well-being at work.

**Figure 4 ijerph-17-06651-f004:**
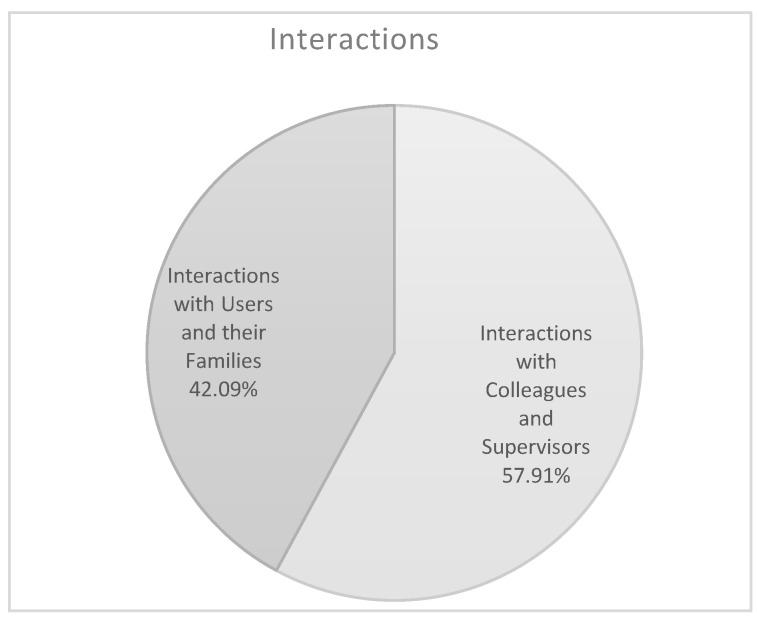
Theme n. 1.

**Figure 5 ijerph-17-06651-f005:**
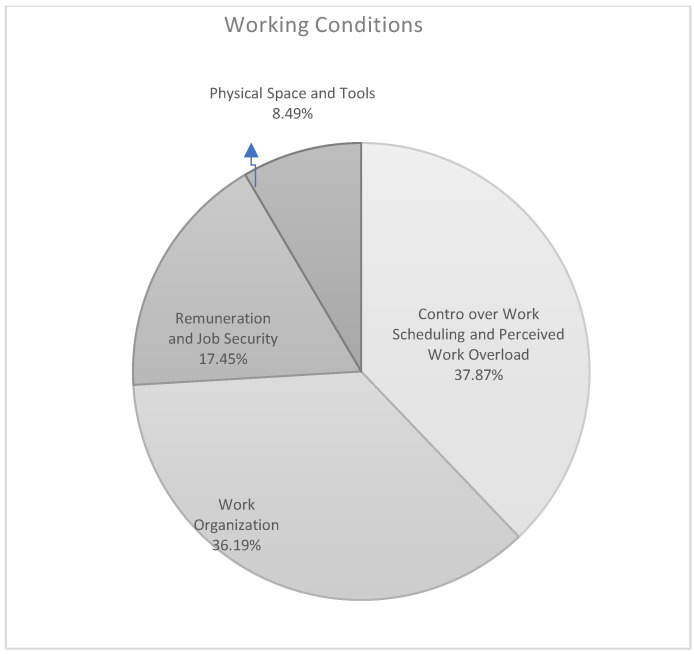
Theme n. 2.

**Figure 6 ijerph-17-06651-f006:**
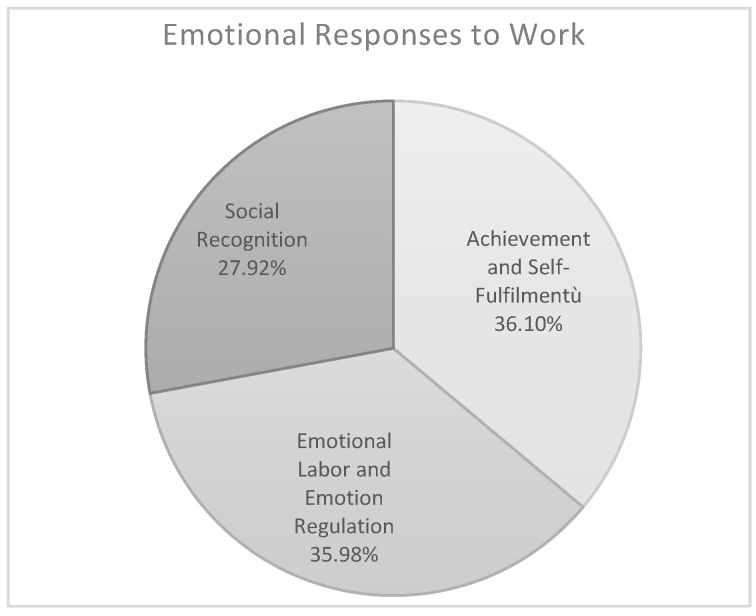
Theme n. 3.

**Figure 7 ijerph-17-06651-f007:**
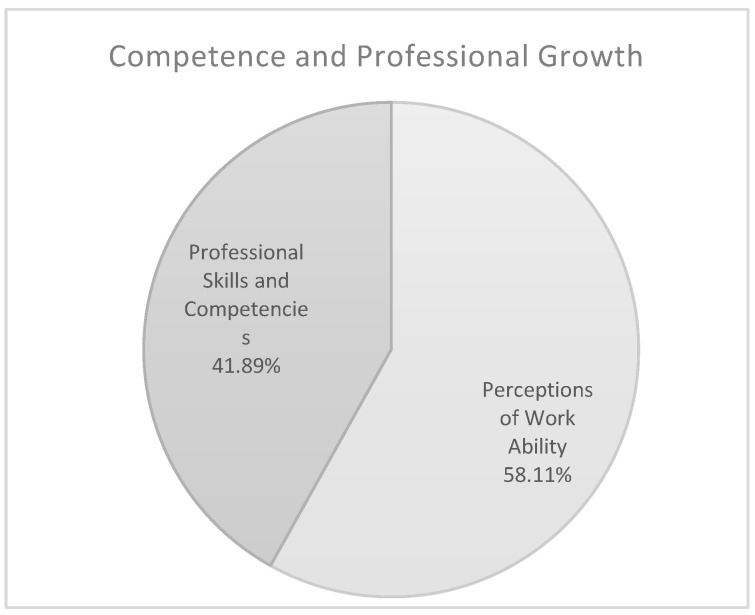
Theme n. 4.

**Table 1 ijerph-17-06651-t001:** Participant characteristics.

Variables	Whole Sample		Educators		Nurses	
**Gender**						
Males	190	(23.9%)	119	(28.1%)	71	(19.1%)
Females	605	(76.1%)	304	(71.9%)	301	(80.9%)
**Age**						
21–30 years	186	(23.4%)	127	(30.0%)	59	(15.9%)
31–40 years	223	(28.1%)	146	(34.5%)	77	(20.7%)
41–50 years	212	(26.7%)	78	(18.4%)	134	(36.0%)
51–60 years	151	(19.0%)	63	(14.9%)	88	(23.7%)
Over 60	18	(2.3%)	4	(0.9%)	14	(3.8%)
Missing value	5	(0.6%)	5	(1.2%)	0	(0%)
**Length of service**						
Up to 10 years	366	(46.0%)	250	(59.1%)	116	(31.2%)
11–20 years	265	(33.4%)	105	(24.8%)	160	(43.0%)
Over 20	156	(19.6%)	60	(14.2%)	96	(25.8%)
Missing value	8	(1.0%)	8	(1.9%)	0	(0%)

**Table 2 ijerph-17-06651-t002:** Risk and protective factors of well-being at work: data comparison between educators and nurses.

Themes and Sub-Themes	Protective Factors of Well-Being			Risk Factors of Well-Being		
	Educators	Nurses	Sig	Educators	Nurses	Sig
**Interactions**	**38.99%**	**49.30%**	*0.019*	**26.47%**	**27.38%**	n.s.
Interactions with colleagues and supervisors	52.90%	51.32%	n.s.	69.53%	66.42%	n.s.
Interactions with users and their families	47.10%	48.68%	n.s.	30.47%	33.58%	*0.027*
**Working conditions**	**14.01%**	**10.30%**	n.s.	**36.91%**	**43.29%**	*0.025*
Control over work scheduling and perceived work overload	35.40%	13.59%	n.s.	42.67%	40.33%	n.s.
Work organization	30.44%	29.13%	n.s.	31.36%	44.63%	n.s.
Remuneration and job security	22.36%	48.54%	n.s.	20.05%	5.49%	n.s.
Physical space and tools	11.80%	8.74%	n.s.	5.91%	9.55%	n.s.
**Emotional responses to work**	**24.80%**	**24.40%**	n.s.	**16.79%**	**21.80%**	n.s.
Achievement and self-fulfillment	75.44%	36.89%	*0.011*	14.69%	0%	n.s.
Emotional labor and emotion regulation	17.19%	49.59%	n.s.	27.68%	52.61%	*0.029*
social recognition	7.37%	13.52%	n.s.	57.63%	47.39%	n.s.
**Competence and professional growth**	**22.19%**	**16.00%**	n.s.	**19.83%**	**7.54%**	*0.011*
Perceptions of work ability	70.20%	68.13%	n.s.	46.89%	26.03%	*0.038*
Professional skills and competencies	29.80%	31.87%	n.s.	53.11%	73.97%	n.s.

Themes (bold).

**Table 3 ijerph-17-06651-t003:** Risk and protective factors of well-being at work: data comparison between younger educators and younger nurses.

Themes and Sub-Themes	Protective Factors of Well-Being			Risk Factors of Well-Being		
	Younger Educators	Younger Nurses	Sig	Younger Educators	Younger Nurses	Sig
**Interactions**	**41.19%**	**44.39%**	**n.s**	**28.88%**	**24.19%**	**n.s**
Interactions with colleagues and supervisors	50.87%	54.40%	n.s	67.66%	58.89%	n.s
Interactions with users and their families	49.13	45.60%	n.s	32.34%	41.11%	0.038
**Working conditions**	**9.40%**	**9.27%**	**n.s**	**37.64%**	**49.19%**	**0.010**
Control over work scheduling and perceived work overload	20.25%	7.89%	n.s	40.46%	40.44%	n.s
Work organization	25.32%	34.21%	n.s	44.27%	44.81%	n.s
Remuneration and job security	45.57%	47.37%	n.s	5.34%	6.01%	n.s
Physical space and tools	8.86%	10.53	n.s	9.92%	8.74	n.s
**Emotional responses to work**	**19.40%**	**24.15%**	**n.s**	**24.71%**	**19.62%**	**n.s**
Achievement and self-fulfillment	43.56%	33.33%	n.s	0.58%	0%	n.s
Emotional labor and emotion regulation	42.94%	54.55%	n.s	58.72%	45.21%	n.s
Social recognition	13.50%	12.12%	n.s	40.70%	54.79%	n.s
**Competence and professional growth**	**30.00%**	**22.20%**	**0.012**	**8.76%**	**6.99%**	**n.s**
Perceptions of work ability	43.65%	41.76%	n.s	29.51%	15.38%	0.016
Professional skills and competencies	56.35%	58.24%	n.s	70.49%	84.62%	n.s

Themes (bold).

**Table 4 ijerph-17-06651-t004:** Risk and protective factors of well-being at work: data comparison between older educators and older nurses.

Themes and Sub-Themes	Protective Factors of Well-Being			Risk Factors of Well-Being		
	Older Educators	Older Nurses	Sig	Older Educators	Older Nurses	Sig
**Interactions**	**37.04%**	**32.42%**	**n.s**	**27.41%**	**26.51%**	**n.s**
Interactions with colleagues and supervisors	51.25%	53.01%	n.s	69.15%	73.42%	n.s
Interactions with users and their families	48.75%	46.99%	n.s	30.85%	26.58%	n.s
**Working conditions**	**12.96%**	**11.59%**	**n.s**	**44.90%**	**34.56%**	**n.s**
Control over work scheduling and perceived work overload	37.50%	32.58%	n.s	55.84%	33.01%	n.s
Work organization	33.93%	30.34%	n.s	22.73%	35.92%	n.s
Remuneration and job security	19.64%	23.60%	n.s	16.23%	24.76%	n.s
Physical space and tools	8.93%	13.48%	n.s	5.19%	6.31%	n.s
**Emotional responses to work**	**21.53%**	**22.14%**	**n.s**	**14.87%**	**15.44%**	**n.s**
Achievement and self-fulfillment	82.80%	70.59%	0.035	15.69%	18.48%	n.s
Emotional labor and emotion regulation	11.83%	20.59%	n.s	29.41%	11.96%	n.s
Social recognition	5.38%	8.82%	n.s	54.90%	69.57%	n.s
**Competence and professional growth**	**28.47%**	**33.85%**	**n.s**	**12.83%**	**23.49%**	**n.s**
Perceptions of work ability	43.09%	43.85%	n.s	47.73%	50.71%	n.s
Professional skills and competencies	56.91%	56.15%	n.s	52.27%	49.29%	n.s

Themes (bold).

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
