# Peer review of "Risk and Protective Factors of Well-Being among Healthcare Staff. A Thematic Analysis"

_ijerph, 2020, doi:10.3390/ijerph17186651_

Round 1
Reviewer 1 Report
Reviewer Comments to Author:
Type of manuscript: Article
Title: Risk and Protective Factors of Well-Being among Healthcare Staff. A Thematic Analysis
Journal: International Journal of Environmental Research and Public Health
Manuscript ID: ijerph-907992
Summary Evaluation: Minor Revisions Required
Summary Comments:
This paper has several positives. It is well-written and brings up several valid points concerning well-being in the healthcare workplace. The study had a very nice sample size evenly distributed between the two categories. I’m not convinced that it fills a gap in the literature as it states in the conclusion. How is it different from other articles? What specifically was that gap that was filled? Also, the data could be explored more by looking at age and years of service categories. The data should also be presented more clearly and fully.
Major revisions:
- Data transparency with more detailed tables/figures
- Further investigation into subcategories within the data
- Further investigation into potential strategies
- Explain the knowledge gap this fills
Minor revisions:
Introduction
Page 2/Line 55 – Remove (
2/66-68 - My understanding was that ward nurses manage a wing or section of nurses. Therefore, their job is more of a supervisor of the nurses compared to regular nurses who interact with patients. This entire paragraph is suggesting more patient interaction than I thought was customary with ward nurses.
93/135/339/355 – ‘wellbeing’ fix to well-being throughout paper.
3/101 – more efficient what? You did not complete your sentence.
3/117-118 – It’s repetitive to have ‘shortage of nurses’ two sentences in a row.
Methods
You could explore the data more. Possibly look at the differences between age groups and the differences between years of service thematic responses; are they statically different comparing the same group educators/nurses (20 yr old nurses vs 20 yr old educators) or are they statistically different between other categories (20s vs 40s, etc.) in nurses, etc.
Results
Figure 3 – This figure could be improved/organized better/improved visually; possibly make pie charts with each theme and name the subthemes/% in each?
Possibly a summary table of the themes/subthemes data comparison between educators/nurses with p-values, percentages/n=?, etc. I assume you only mentioned the differences between subthemes when they were statistically different between educators and nurses, but it would be nice to have all the data in a table.
Discussion
8/311 – Did you mean work organization instead of working conditions? Sentence is confusing.
9/360 – You’ve provided the areas, but only two potential strategies. I think you could further develop this section.
9/361 – I would reword ‘last worker’s population’.
9/363 – Are you referring to all comparisons between educators and nurses with ‘above mentioned analogies’, or something specific in mind?
9/364 – Are you suggesting they should be paid similarly? The education/qualification to be a nurse has to be more extensive than an educator, why should they be paid similarly? Also, this concept that the educators feel they don’t make enough money is not included in the results.
9/368 – Whatever literature you are referring to here should be referenced again.
Conclusions
10/383 – Not clear how this fills a gap in knowledge. It seems as if you adequately demonstrated how wide of a base of knowledge there is on this topic.
Author Response
Dear Editor,
Dear Reviewer 1,
Dear Reviewer 2,
We would like to thank you and the reviewers for your helpful comments. We truly believe that your suggestions improved the quality of our paper.
We have revised the paper according to the reviewers’ comments. Below we reported our replies to the Reviewers.
We have also corrected the format of references cited in the main text.
We confirm the ethic code. The research process followed the ethical guidelines of the Italian Psychological Association (AIP) and got ethical approval by the Ethic Committee of our Department, reference number 284-77324 on date 26/11/2014.
Point: Page 2/Line 55 - Remove (
Response Line 55 - We have removed the parenthesis.
Point: 2/66-68 - My understanding was that ward nurses manage a wing or section of nurses. Therefore, their job is more of a supervisor of the nurses compared to regular nurses who interact with patients. This entire paragraph is suggesting more patient interaction than I thought was customary with ward nurses.
Response: As “ward nurses” we mean nurses working in a hospital ward; in Italy, they can be either “simple” nurse or responsible for the coordination of all the nurses of the ward (caposala or coordinatore infermieristico). In this case, one of their responsibilities is to manage possible conflicts with the patients and their families. However, even in the role of coordinator (which is not a supervisor), ward nurses are usually physically in the ward and involved in the interaction with patients and their families.
Point: 3/101 – more efficient what? You did not complete your sentence.
Response: 3/101 Now Line 100 - We have completed our sentence.
Point: 3/117-118 – It’s repetitive to have ‘shortage of nurses’ two sentences in a row.
Response: 3/117-118 Now Lines 113-115 - We have deleted the repetitive sentence.
Points:
Data transparency with more detailed tables/figures
Further investigation into subcategories within the data
Data could be explored more by looking at age and years of service categories. The data should also be presented more clearly and fully. You could explore the data more. Possibly look at the differences between age groups and the differences between years of service thematic responses; are they statically different comparing the same group educators/nurses (20 yr old nurses vs 20 yr old educators) or are they statistically different between other categories (20s vs 40s, etc.) in nurses, etc.
Possibly a summary table of the themes/subthemes data comparison between educators/nurses with pvalues, percentages/n=?, etc. I assume you only mentioned the differences between subthemes when they were statistically different between educators and nurses, but it would be nice to have all the data in a table.
Figure 3 – This figure could be improved/organized better/improved visually; possibly make pie charts with each theme and name the subthemes/% in each?
Response: In the "Results" Section, according to Reviever 1 we have presented the data more clearly and fully. In the previous manuscript version in the text we have only mentioned the statistically differences between educators and nurses. According to Reviewer 1 we have added a summary table of the themes/subthemes data comparison between educators and nurses with pvalues and percentages. We have added more detailed tables (Table 2, Table 3 and Table 4). According to Reviewer 1 we have changed into pie charts Figure 3 (now Figure 3 to 7). We have also conducted further investigations to explore the data more. We have explored the differences between age groups thematic responses, comparing the same group educators/nurses (young and old nurses versus young and old educators), even if there are only few differences (Table 3 and Table 4).
Points:
Further investigation into potential strategies
9/360 – You’ve provided the areas, but only two potential strategies. I think you could further develop this section.
Response: 9/360 Now Lines 416-436 - We have added further potential strategies to improve well-being at work.
Point: 8/311 – Did you mean work organization instead of working conditions? Sentence is confusing.
Response: 8/311 Now Lines 360-363 - We have changed the sentence.
Points:
9/361 – I would reword ‘last worker’s population’.
9/363 – Are you referring to all comparisons between educators and nurses with ‘above mentioned analogies’, or something specific in mind?
9/364 – Are you suggesting they should be paid similarly? The education/qualification to be a nurse has to be more extensive than an educator, why should they be paid similarly? Also, this concept that the educators feel they don’t make enough money is not included in the results.
Response: Lines 361, 363 and 364 - We have removed these sentences.
Point: 9/368 – Whatever literature you are referring to here should be referenced again.
Response: 9/368 Now Line 435 - We have added the referenced literature.
Points:
I’m not convinced that it fills a gap in the literature as it states in the conclusion. How is it different from other articles? What specifically was that gap that was filled?
10/383 – Not clear how this fills a gap in knowledge. It seems as if you adequately demonstrated how wide of a base of knowledge there is on this topic.
Explain the knowledge gap this fills
Response: 10/383 Now Lines 447-454 – We have changed the Conclusion, and we have explained the knowledge gap this fills.
Reviewer 2 Report
I thank you for the work you have done. The topic is interesting, and the data set is original, but the method section needs a significant improvement, which affects the discussions. I recommend a revision, and I offer my suggestions to help the authors to improve their article.
[1]
Page 3, the authors wrote:
“Given that well-being has been found to be related to performance in the work setting, it is not surprising that the concept of well-being has attracted much attention [32]”
- I believe more citations are needed to support the above claim: “has attracted much attention”.
[2]
Related to this, the authors ended the paragraph by claiming that “little attention has been paid to the healthcare staff’s experience of well-being”.
- I believe this claim warrants to be substantiated. To this end, please see the publication outputs on healthcare staff’s experience of well-being below.
Houghton, C., Murphy, K., Brooker, D., & Casey, D. (2016). Healthcare staffs’ experiences and perceptions of caring for people with dementia in the acute setting: Qualitative evidence synthesis. International Journal of Nursing Studies, 61, 104-116.
Bojner Horwitz, E., Grape Viding, C., Rydwik, E., & Huss, E. (2017). Arts as an ecological method to enhance quality of work experience of healthcare staff: a phenomenological-hermeneutic study. International journal of qualitative studies on health and well-being, 12(1), 1333898.
[3]
The authors wrote:
All over the world, nurses are experiencing greater workloads as a result of rising average patient acuity, fewer support resources and a shortage of nurses [37]
- Citation [37] is an Australian based work, and by implication, Australia (or at best Western societies) cannot represent "all over the world."
[4]
Concerning the overall presentation of the article, improvement is required. For instance, improvement is needed in the area of paragraphing. For example, a stand-alone sentence below is not appropriate to represent a whole paragraph.
“In conclusion, burnout and turnover among healthcare staff are not just an organizational challenge or a topic for economic analysis but a global issue with real-world effects in terms of quality of health care [23,46,47].”
[5]
The authors "adopt a qualitative approach without a pre-existing theoretical framework" and data was thematically analysed. This is problematic, and consequent to this feature, the method section needs a significant improvement. The authors can overcome this limitation by following one of the following roads.
- Route one:
I do not believe that the authors enter the field tabula rasa, i.e., unencumbered by notions of the phenomena we seek to understand. As a result, theory (i.e., a theoretical framework), then, includes any general set of ideas that guide action, and that theory profoundly affects the conduct of qualitative research. So, I advise the authors to adopt a suitable theoretical framework of their choice in revising their article. This is because it aligns with the thematic analytic approach you have chosen. A theoretical framework guides us, whether it is made explicit or not.
- Route two:
Alternatively, should the authors believe still that a theoretical framework is not needed, they should:
Give some rationales for adopting a qualitative approach without a pre-existing theoretical framework". Take the "Grounded Theory" route, which is more a data-driven route than the thematic approach. "Grounded theory" holds that qualitative studies can be done to discover theory instead of relying on a pre-existing theoretical framework. Here, the researchers start by collecting data and then searches for theoretical constructs, themes, "thematic map", that is, patterns that are primarily data-driven.
Depending on the route the authors choose to take in revising their article, they should also amend the discussion section to align with the method section.
I look forward to reading your revised version, and I hope you found my suggestions to be useful.
Author Response
Dear Editor,
Dear Reviewer 1,
Dear Reviewer 2,
We would like to thank you and the reviewers for your helpful comments. We truly believe that your suggestions improved the quality of our paper.
We have revised the paper according to the reviewers’ comments. Below we reported our replies to the Reviewers.
We have also corrected the format of references cited in the main text.
We confirm the ethic code. The research process followed the ethical guidelines of the Italian Psychological Association (AIP) and got ethical approval by the Ethic Committee of our Department, reference number 284-77324 on date 26/11/2014.
Point1: Page 3 - I believe more citations are needed to support the above claim: “has attracted much attention”.
Response: Page 3 Line 88 – We have added more citations.
Point 2: I believe this claim warrants to be substantiated.
Response: Lines 88-91 – We have changed the claim.
Point 3: Citation [37] is an Australian based work, and by implication, Australia (or at best Western societies) cannot represent "all over the world."
Response: Lines 113-115 – We have deleted the sentence. We have added more citations to support the importance of the two problems identified by international literature.
Point 4: Concerning the overall presentation of the article, improvement is required. For instance, improvement is needed in the area of paragraphing. For example, a stand-alone sentence below is not appropriate to represent a whole paragraph.
Response: We have improved the area of paragraphing in all the manuscript.
Point 5: The authors "adopt a qualitative approach without a pre-existing theoretical framework" and data was thematically analysed. This is problematic, and consequent to this feature, the method section needs a significant improvement. The authors can overcome this limitation by following one of the following roads.
Response: Thank you for your punctual and rich comments, aiming to improve the methodological section of this work.
As a research group, we feel in agreement with Boyatzis’ conceptualisation (1998) of thematic analysis as “bridge” or “translator” between two different epistemologies, the positivist underlying the quantitative methodologies, and the postmodern/interpretative ones. Accordingly, “thematic analysis is not another qualitative method but a process […] for encoding qualitative information” (Boyatzis, 1998:4).
Boyatzis, R. E. (1998). Transforming qualitative information: Thematic analysis and code development. Thousand Oaks, CA: Sage.
We rejected the possibility to follow the “route two” suggested in the review, since it does not correspond to the approach we applied. At the same time, we are aware that conceptualisiations around thematic analysis developed over time giving rise to a lively and stimulating debate (just to mention some authors, see: Braun & Clarke, 2013; 2019; Braun, Clarke, Hayfield & Terry, 2019).
Braun, V., & Clarke, V. (2013). Successful qualitative research: A practical guide for beginners. London: Sage.
Braun, V., & Clarke, V. (2019). Reflecting on reflexive thematic analysis. Qualitative Research in Sport, Exercise and Health, 11(4), 589-597.
Braun, V.; Clarke, V.; Hayfield, N.; Terry, G. Thematic Analysis. In Handbook of Research Methods in Health Social Sciences, Liamputtong, P. Ed.; Springer: Singapore, 2019, pp. 843-860.
Epistemological issues are crucial in such debate, since thematic analysis appears as an umbrella term: researchers need to clarify their own theoretical assumptions, that necessarily affect the methodological procedure.
Your comments pinpointed this lack of attention in the article, and we are sincerely grateful to have made it explicit. We have better articulated the methodological assumption underlying our choice of the thematic analysis, as described in lines (see: paragraph 2 "Materials and Method" Lines 140-156; and paragraph 2.3 "Data Analysis" Lines 191-218). We hope to have framed more precisely the methodological procedure of the research now, and in doing so to have been able to enhance the methodological section.
Round 2
Reviewer 2 Report
I thank you very much for revising your article, and I am glad you found my recommendations to be useful.
You have successfully addressed all my previous concerns [that is, number 1,3,4 & 5]. However, my point number [2] has not yet been adequately addressed.
In particular, I invited you to have a look at two relevant pieces of work I read while reviewing your paper. I was expecting you to report back to me about what you found about at least one of the articles I mentioned. Whether you decide in the end to use what you found in your work or not is completely up to you. But it is critical for the reviewer and the authors to have a full and engaged dialogue about matters that concern the manuscript in submission. In this regard, you have missed commenting on point number 2 (see my previous review report). I believe it relates to the topic of your inquiry - the gap - about healthcare staff’s experience of well-being.
I look forward to hearing from you.
Author Response
Dear Reviewer
We would like to thank you for your helpful comment. We have revised the paper according to your comment (point number 2).
Point: You have successfully addressed all my previous concerns [that is, number 1,3,4 & 5]. However, my point number [2] has not yet been adequately addressed.
In particular, I invited you to have a look at two relevant pieces of work I read while reviewing your paper. I was expecting you to report back to me about what you found about at least one of the articles I mentioned. Whether you decide in the end to use what you found in your work or not is completely up to you. But it is critical for the reviewer and the authors to have a full and engaged dialogue about matters that concern the manuscript in submission. In this regard, you have missed commenting on point number 2 (see my previous review report). I believe it relates to the topic of your inquiry - the gap - about healthcare staff’s experience of well-being.
Response: Thank you for your pinpointing this references, actually we had missed this point. We have read the articles and inserted the quotes where relevant (Lines 89-93 and Lines 424-426).
